# In Vitro Skin Models as Non-Animal Methods for Dermal Drug Development and Safety Assessment

**DOI:** 10.3390/pharmaceutics17101342

**Published:** 2025-10-17

**Authors:** Viviana Stephanie Costa Gagosian, Raquel Coronel, Bruna Caroline Buss, Maria Luiza Ferreira dos Santos, Isabel Liste, Berta Anta, Leonardo Foti

**Affiliations:** 1Graduate Program in Bioscience and Biotechnology, Carlos Chagas Institute (ICC/FIOCRUZ-PR), Curitiba 81310-020, PR, Brazil; 2Unidad de Regeneración Neural, Unidad Funcional de Investigación de Enfermedades Crónicas (UFIEC), Instituto de Salud Carlos III (ISCIII), 28220 Majadahonda, Madrid, Spain; 3Graduate Program in Biological Sciences, Federal University of Paraná, Curitiba 80060-000, PR, Brazil; 4Unidad de Biología Celular, Unidad Funcional de Investigación de Enfermedades Crónicas (UFIEC), Instituto de Salud Carlos III (ISCIII), 28220 Majadahonda, Madrid, Spain; 5Laboratory of Molecular Biology and Trypanosomatids Systemic, Carlos Chagas Institute (ICC/FIOCRUZ-PR), Curitiba 81310-020, PR, Brazil

**Keywords:** reconstructed human skin models, toxicology, regulations, tissue engineering, topical delivery, transdermal delivery, new approach methodologies (NAMs)

## Abstract

**Highlights:**

**What are the main findings?**

**What is the implication of the main finding?**

**Abstract:**

Research on in vitro skin models has advanced remarkably, driven by a better understanding of the skin and the search for more ethical and efficient methods. The development of these models was initially motivated by the need for reduced animal testing and a faster and more ethical approach for the safety evaluation of cosmetic and pharmaceutical products. Stricter regulations and growing ethical awareness have driven further evolution, resulting in more refined and reliable methods. Diversity of cell types is crucial to replicating the complexity of human skin, including epithelial, dendritic, endothelial, and adipose cells, providing environments that closely mimic the physiological skin environment. This allows for more precise studies on skin interactions with cosmetic, dermatological, and pharmaceutical products. In vitro skin models have applications in toxicity testing, dermatological product evaluation, skin ageing studies, and drug research, reducing dependence on animal testing. This review presents a look at the different types of in vitro skin models developed for various applications, with a brief look at their strengths and drawbacks. Models developed for disease-specific applications are also covered. Techniques such as bioprinting and organ-on-a-chip have revolutionised the manufacturing of these models. Challenges persist, such as the need to improve vascularisation and faithfully replicate skin architecture. The promising future of these models points to an exciting path forward for dermatological research and the cosmetic industry. This review addresses the history and regulations of skin models, explores various skin models, and highlights the most recent advances, outlining future perspectives and offering a comprehensive overview.

## 1. Introduction

The skin is the largest organ in the body and performs several vital functions. One of its main tasks is to protect against external insults, including physical, chemical, or biological threats. This is the first line of defence of the body and plays a crucial role in the immune system. Furthermore, the skin actively contributes to temperature regulation, preventing excessive water loss from the body [1].

Skin exhibits an intricate structure and is composed of two fundamental layers, the dermis and the epidermis. The epidermis, the outermost layer, is a non-vascularized stratified epithelium, predominantly made up of ectoderm-derived keratinocytes, that undergoes continuous cell renewal [2]. These cells play a crucial role in the defence functions of the skin, controlling inflammation through the release of pro-inflammatory cytokines essential for modulating skin irritation and sensitisation [3,4]. Additionally, the appendages (hair follicles, sebaceous glands, and sweat glands) are located in this layer. Moreover, the keratinisation process, which involves cell differentiation into several layers (basal layer, stratum spinosum, stratum granulosum, stratum lucidum, and stratum corneum), is crucial in constituting an essential barrier against the penetration of chemical substances and microorganisms [2,5]. The dermis is connected to the epidermis through the basement membrane, constituting connective tissue. This stratum contains sweat glands, hair follicles, muscles, sensory neurons, and blood vessels, with fibroblasts being the main cell type identified [2,6] (Figure 1).

As skin comes in direct contact with the environment, it is exposed to various chemical and physical agents capable of causing damage or triggering adverse effects [7,8]. According to Pesonen et al. (2021), hairdressers and cosmetologists have a high rate of occupational skin diseases due to frequent contact with products used in their professions, leading to issues such as skin irritation and dermatitis, often affecting their hands [9]. Allergic contact dermatitis is more prevalent than irritant contact dermatitis, being associated with exposure to hair care products, including detergents, additives, permanent wave solutions, bleaching agents, fragrances, dyes, nail acrylates, and nickel sulphate (NiSO_4_) in cosmetology equipment [9,10]. Therefore, it is crucial to identify these agents in terms of their skin irritation potential, considering the different skin types present in individuals, including those with and without dermatological conditions [9,10].

Experiments involving animals have been recorded as early as the 5th century BC; however, their use has increased significantly since the 19th century [11]. Several animal species have been widely used in research aimed at the development and risk assessment of medicines, cosmetic products, chemical substances, medical devices, and vaccines, consolidating themselves as important models for understanding the effects resulting from the use of or exposure to these products [12,13]. It is estimated that approximately 200 million animals are currently used annually in scientific research, giving rise to ethical and scientific concerns [14]. The main limitations of these models include the difficulty in extrapolating the data obtained to humans owing to the unique physiological characteristics and biological complexity of different species, which can compromise the predictability of toxic effects and pathological conditions that may occur in humans [15,16].

Therefore, several alternatives to the use of animals in experiments have been proposed, especially in drug and chemical testing. These methods offer advantages such as greater time efficiency and cost effectiveness and lower labour requirement, in addition to being aligned with the ethical principles of scientific research [12,17].

Two-dimensional models, characterised by monolayer growth, are widely used in various assays to understand biological processes. However, these models have limitations as they lose essential phenotypic and functional characteristics, such as the cell differentiation in the epidermis, because of dissimilarities with the in vivo environment. In contrast, 3D culture models have emerged as complex systems that reproduce the morphological and functional characteristics of in vivo tissues and can be monotypic (using a single cell type) or multicellular (co-culture) [18,19].

Three-dimensional (3D) skin models outperform 2D models by preserving the tissue architecture, nutrient/oxygen gradients, and cell–cell/matrix signalling that regulate epidermal differentiation, stratum corneum formation, and inflammatory responses. These aspects are lost in 2D monolayers and result in less predictable gene expression and pharmacodynamic profiles in humans [18,20]. This can be seen in reconstructed human epidermis (RHE) and human skin equivalents (HSE), which better reproduce the skin barrier and allow functional readings such as transepithelial/epidermal electrical resistance (TEER) and transepidermal water loss (TEWL). These characteristics allow for skin-representative topical assays, in accordance with the RHE method for skin irritation (OECD TG 439) [21].

Platforms such as skin-on-a-chip (SoC) add controlled perfusion and shear, facilitate co-cultures with endothelial and immune components, and extend the experimental windows for permeation, inflammation, and metabolism, bringing the microenvironment closer to in vivo conditions and strengthening in vitro–in vivo correlation (IVIVC) in cutaneous pharmacology [22,23].

In 2024, the European Medicines Agency (EMA) updated its guideline on quality and equivalence for locally acting dermal products, emphasising in vitro assays such as the in vitro release test (IVRT) and the in vitro permeation test (IVPT), in addition to skin integrity criteria. This framework consolidates RHE, HSE, and SoC platforms as new approach methodologies, or NAMs, for the development and evaluation of dermal and transdermal products [24,25]. In 2022, the Food and Drug Administration (FDA), established these platforms as support for the screening, classification, and IVPT of formulations, with barrier and integrity readouts such as TEER, TEWL, and tritiated water (HTO) [26].

This review presents a comprehensive analysis of current in vitro skin models, focusing on several areas of study, including dermatology, cosmetics, pharmaceuticals, and toxicology. A detailed understanding of the structure and functions of the skin is essential for significant advances in these areas. This review critically addresses regulations and the types and sources of cells used, as well as explores models that replicate specific pathological conditions. It is important to develop a thorough understanding of these models, highlighting their applicability and innovative potential in research and development in health sciences and dermatology.

For in vitro skin permeation studies with human skin, the European Medicines Agency (EMA) guideline for equivalence of locally acting topical products in the European Union recommends including at least 12 skin donors, with 2 or more replicates per donor, totalling at least 24 experiments per product. Equivalence should be demonstrated by 90% confidence intervals within 80–125% for both peak permeation flux and total permeate quantity. In cases of high variability, clinically justified widening may be applied, accepting ranges of 69.84–143.19% [24]. In parallel, the Organisation for Economic Co-operation and Development (OECD) Test Guideline TG 439 provides a validated RHE method for skin irritation hazard identification that underpins non-animal safety assessment [21]. While these tools can support in vitro–in vivo correlation (IVIVC) for selected products, ongoing efforts to improve HSE barrier properties and to integrate dynamic perfusion in SoC aim to further narrow gaps to native skin and to strengthen decision-making in topical and transdermal development [23].

### Methods of the Narrative Review

This is a narrative (non-systematic) review. We searched PubMed, Scopus and Web of Science (January 2018–August 2025) using predefined terms related to in vitro skin models (RHE/HSE, immunocompetent, vascularised/adipose-integrated, appendages, skin-on-a-chip, bioprinting) and pharmaceutical endpoints (barrier, TEER, permeation, irritation/sensitisation). We prioritised peer-reviewed articles from the last seven years, adding seminal/regulatory sources (e.g., OECD) as needed. Inclusion: human cell-based models (primary, immortalised, iPSC) and ex vivo explants relevant to dermal/transdermal drug development and safety. Exclusion: animal-only models and sources lacking methodological detail. We extracted model attributes (cell types/sources, structure, immune/endothelial/adipose/appendage components, culture platform), functional readouts, validation/regulatory status and applications. No protocol registration, risk-of-bias assessment or meta-analysis was performed.

This manuscript uses ChatGPT version 5 Pro (OpenAI) to improve the clarity, cohesion, and grammar of the text. The tool was used to clarify linguistic and stylistic adjustments, without making changes to the scientific content or conclusions. All final decisions regarding the writing and structure of the manuscript were made entirely by the authors.

## 2. History and Regulation

The use of animals in scientific and medical research has been a common practice throughout history and has contributed to significant advances in various fields. However, there is growing concern among researchers that animal experimentation may not always be based on scientifically sound premises, and its continued acceptability is a result of the lack of better alternatives [15].

Although there are notable genetic, physiological, cellular, and molecular similarities between several animal models and humans, it is crucial to recognise the limitations inherent to the use of animals, and the extrapolation of results from models to humans must be carried out with caution. Therefore, increasingly, scientific organisations and government regulatory agencies recognise that alternative methods have the potential to replace animal testing, contributing to improved efficiency and safety in the development of new treatments for humans [15,27].

The practice of testing pharmaceutical products on animals began in 1937 in the United States following a tragic incident that resulted in the death of 107 people, including adults and children [15]. This event led to the establishment of the U.S. Federal Food, Drug, and Cosmetic Act in 1938, which made toxicity testing on animals mandatory [28,29,30]. Subsequently, in 1946, the requirement was incorporated into the Nuremberg Code (National Institutes of Health, 1949) and later into the Declaration of Helsinki in 1964 (World Medical Assembly Declaration of Helsinki, 1964), establishing that clinical trials involving human participants should be planned based on results obtained through prior animal experimentation [15].

In 1959, Russell and Burch introduced the 3Rs principle—Replace, Reduce, and Refine—which aimed to promote the ethical use of animals by encouraging alternative in vitro methods for testing products with toxic potential [31].

In Europe, the regulation of cosmetic products began in 1976 when member countries of the European Economic Community (EEC) adopted Directive 76/768/EEC. This directive aimed to harmonise cosmetic regulations by establishing common criteria for safety, labelling, and packaging [32,33].

In 1997, the United States took additional steps toward reducing animal testing with the creation of the Interagency Coordinating Committee for the Validation of Alternative Methods (ICCVAM) and the National Toxicology Program Interagency Center for the Evaluation of Alternative Toxicological Methods. These agencies were tasked with coordinating the development, validation, and promotion of alternative toxicology testing methods across U.S. government institutions [34]. The FDA emphasised that, for cosmetic testing purposes, alternative scientifically valid methods should be considered before resorting to whole-animal testing [35].

In the early 2000s, significant progress was made toward alternative testing approaches. In 2000, a pre-validation study of the first in vitro skin corrosion model using 3D reconstructed human skin models was published, offering an alternative to animal testing [36,37]. By 2002, this method had been validated by the Organization for Economic Co-operation and Development (OECD) as the first official alternative for cosmetic product safety assessment [38].

Building on these efforts, Directive 2003/15/EC (the Seventh Amendment to Directive 76/768/EEC) scheduled a ban on animal testing for finished cosmetic products starting in 2004. This ban was widened in 2009 to include cosmetic ingredients. However, products tested on animals outside of Europe could still be marketed within the European Union. A full ban on animal testing for cosmetic purposes came into effect only in 2013, following a 20-year process supported by extensive investment [33].

Regulation of the use of non-animal skin models varies from country to country regarding the scope of accepted methods, the role of defined approaches, and the pace of updating technical standards. In some countries, these methods are already a starting point for safety assessment, but in others, their adoption is gradual and conditional on product categories or international standards. In the European Union, in vitro RHE methods for skin irritation and corrosion are widely accepted, in particular OECD TG 439 and related standards. The European Medicines Agency (EMA) has begun to emphasise the in vitro release test (IVRT) and the in vitro permeation test (IVPT) in its quality and equivalence guidelines for local skin products, which has reinforced the adoption of these models in regulatory dossiers. In parallel, the European Chemicals Agency (ECHA), within the scope of REACH, advises registrants on the use of OECD in chemico and in vitro guidelines and defined approaches (DAs) for skin sensitization. In the case of medical devices, the guidelines are being updated, keeping biocompatibility assessment aligned with revisions to the ISO 10993 series [39,40,41,42].

In the United States, the Food and Drug Administration published specific guidance for IVPT testing in abbreviated new drug applications (ANDAs) for topical drugs, consolidating the routine use of in vitro human skin and barrier integrity readings in equivalence strategies [26,43,44]. In the United Kingdom, the government confirmed in 2023 that it will no longer issue licences for animal testing of substances used exclusively as cosmetic ingredients, in alignment with validated and internationally accepted alternatives. Under the UK REACH, the Health and Safety Executive (HSE) clarifies that the defined approaches (DAs) of the OECD TG 497 guideline can fully meet the skin sensitization requirements, including potency estimation [45,46].

In Canada, the ban on animal testing for cosmetics came into effect on 2023, with the publication of implementation guidelines for industry, prompting a migration to alternative methods such as reconstructed skin models [47,48,49].

Since July 2016, in Brazil, the National Health Surveillance Agency (ANVISA) has established standards to reduce the use of animals in safety and efficacy testing of medicines, cosmetics, health products, sanitizers, and related products, mandating the priority adoption of alternative methods recognised by the National Council for the Control of Animal Experimentation (CONCEA), whenever available [50]. In addition to the formal recognition of methods by CONCEA and the creation of BRACVAM, there was regulatory progress with Normative Resolution CONCEA 58 of 2023, which prohibits the use of animals when safety and efficacy have already been demonstrated by validated methods. Recent legislative initiatives aimed at a broader prohibition in the cosmetics sector has accelerated the adoption of RHE and related testing in regulatory practice [50,51,52].

In China, since 2021, certain general-use cosmetics can obtain exemptions from animal testing based on regulatory criteria, paving the way for greater use of in vitro methods, although additional requirements remain for special categories. In 2024, the NIFDC published technical guidelines that include integrated strategies for skin sensitization testing and assessment, expanding the regulatory use of ADs [53].

The ban on animal testing for cosmetics and the importation of cosmetics tested on animals consolidated the transition to alternatives, including non-animal skin models was achieved in India in 2013 [54].

Regulatory reforms in South Korea culminated in measures that penalise the use of animal testing for cosmetics and promote its replacement with internationally accepted alternative methods [55,56].

The Japanese regulatory agency (JaCVAM) supported the national adoption of OECD guidelines, including TG 439, which allows the regulatory use of reconstructed epidermis models in skin irritation and corrosion assessments, and a 2025 statement recognising the regulatory utility of a DA for skin sensitization, in line with the national adoption of OECD methods [57].

Regulations on cosmetic testing are well established and harmonised internationally, with widespread acceptance of non-animal methods and defined approaches for skin irritation, corrosion, and sensitization. In the case of pharmaceutical products, especially systemic ones, and even topical ones, the regulatory landscape remains heterogeneous, requirements vary between agencies, the validation of alternative methodologies is progressing unevenly, and many decisions are still based on clinical data and, in some cases, traditional preclinical evidence [58,59].

Despite regulatory progress, many experiments involving animals are still conducted today, particularly for immunotoxicity testing. These studies face numerous challenges, including high costs, ethical concerns, and questionable relevance for human risk assessment. Therefore, researchers continue to investigate and validate alternative in vitro methods capable of predicting the toxic potential of cosmetic ingredients [60,61]. However, comprehensive strategies to assess the predictability and reproducibility of testing approaches for specific regulatory needs are yet to be developed. This constitutes a major obstacle to their regulatory approval and implementation of novel in vitro methods, ultimately leading to limited confidence in their application [60,62].

The FDA and U.S. lawmakers have acknowledged the inherent limitations of animal testing and, by passing the FDA Modernization Act 2.0 in December 2022 [43,44], authorised the use of more accurate and human-relevant alternative methods for preclinical drug evaluation. These alternatives include technologies such as induced pluripotent stem cells (iPSCs), organoids, organs-on-chips (OoC), and advanced artificial intelligence tools, including generative adversarial networks (GANs), large language models, and synthetic digital twins [63,64]. By integrating multiomics and structural biology data, these cutting-edge approaches offer enhanced predictive capabilities for assessing toxicity, metabolism, and therapeutic efficacy in humans [65]. This legislation marks a significant paradigm shift, steering preclinical research away from outdated animal models and toward innovative, humane, and scientifically advanced methodologies [66].

Updates over the last three years on in vitro skin models, organised by regulatory agencies in each country, are presented in Table 1, which indicates the accepted method or model, the year and the intended purpose (Table 1).

## 3. Cell Types Used in In Vitro Skin Models

Many of the skin models reported in the literature are constructed with primary cells collected from surgically discarded tissue. This strategy ensures a source of quality cells while minimising the need for invasive collection [69]. However, the importance of ethical and regulatory considerations in the use of these tissues is worth highlighting, wherein the use of any material of biological origin from donors is a process subject to review and approval by a Research Ethics Committee. Additionally, obtaining informed consent from the donor while fully adhering to ethical principles and relevant regulations is essential. These measures guarantee ethical and legal compliance in the conduct of research, ensuring protection of the rights and well-being of donors [70,71].

One of the main objectives of developing alternative methods to animal models is to mimic human morphology, physiology, and function, thereby creating a setting comparable to the in vivo environment to obtain authentic responses in experiments [17]. Primary cells are favoured for their superior ability to preserve the characteristics of the organism in vivo compared to established cell lines. These cells maintain their phenotypic and functional characteristics, providing a more accurate representation of the biological complexity of the original tissue [72].

Primary cells provide crucial authenticity in studies aiming to replicate the physiological conditions of the skin, particularly in the dermatology, toxicology, and cosmetics fields. However, using primary cells poses challenges because of high variability among donors, which makes reproducibility between experiments difficult. Moreover, these cells can only be used for a few passages, necessitating frequent collection, which further presents a challenge, given the limited availability of donor tissue [72,73,74].

Models prepared using commercially available lineage cells offer a viable alternative, resembling the morphology and function of skin in vivo without involving ethical or logistical concerns. Furthermore, these cell lines can be maintained in culture for multiple passages, eliminating variability between donors and providing a longer-lasting source of cells, allowing numerous tests to be conducted with the same cell line [75].

However, it is important to highlight that, despite their advantages in terms of durability and practicality, immortalised cells cannot replicate all in vivo physiological characteristics. This limitation may make them less reliable for certain experiments, especially those that require a more accurate representation of the physiological conditions of the skin [76].

Models built with human pluripotent stem cells (hPSC), which include human embryonic stem cells (hESCs) and human-induced pluripotent stem cells (hiPSCs), can potentially overcome challenges associated with models constructed using cell lines and primary cells, especially in studies of the pathophysiology of human skin diseases. hPSCs exhibit a remarkable capacity for self-renewal and differentiation into all cell types derived from the three embryonic germ layers, making it possible to build several models from a single donor, thereby offering a virtually unlimited resource and the ability to generate genetically identical skin models, reproducing skin morphology, physiology, and function with low genetic variation [77,78,79,80,81].

It is important to highlight that 3D skin cultivation using hPSC also presents challenges, especially related to handling and the cost of the technology. The time required for maintenance and differentiation of hPSCs can be prolonged, requiring specialised handling skills and significant investments compared with conventional cell sources. However, these complexities are largely related to the initial establishment of the hPSC line. Once established, handling and cost are similar to those of other cell types. Additionally, the existence of cell banks that offer hPSCs and iPSCs can facilitate the development of these models, especially in areas such as drug development and investigation of new therapies [81].

The choice between primary cells, established cell lines, or hPSC to construct skin models will depend on the specific goals of the experiment as well as the desired characteristics of the in vitro model, including the required cell types. Each approach has its advantages and limitations, highlighting the importance of a careful choice to ensure the relevance and authenticity of the model developed [82,83].

Interestingly, 3D hPSC skin models have been developed and subsequently undergone genome editing to express specific functional, biochemical, or pathological characteristics. This has been achieved through gene editing techniques, such as the CRISPR/Cas9 system, or by deriving skin lines that exhibit mosaicism. These approaches have been explored by several authors, as highlighted by Marinova et al. (2021) [84], Dabelsteen et al. (2020) [85], and Enjalbert et al. (2020) [86]. Marinova et al. (2021) [84] utilised the CRISPR-Cas9 system to edit the gene POFUT1, which is involved in glycosylation. The primary goal was to generate 3D organotypic skin models using genetically edited human keratinocytes (N/TERT-1 line) to study the role of glycans in epithelial tissue formation and homeostasis. Although this research does not address a specific disease directly, these models have potential applications in understanding diseases related to glycosylation disorders. Similarly, Dabelsteen et al. (2020) [85] developed a human organotypic platform based on genetically edited cells using CRISPR-Cas9. Their aim was the systematic and detailed analysis of specific glycan structures in tissue development and maintenance. Although not directly linked to a specific disease, the findings provide groundwork for future studies on conditions involving altered glycosylation, such as congenital disorders of glycosylation. Enjalbert et al. (2020) [86] employed CRISPR/Cas9 gene editing to generate a knockout (KO) cell line for the ABCA12 gene to investigate the pathological mechanisms of Harlequin Ichthyosis (HI), a severe skin disease caused by loss-of-function mutations in ABCA12. This study identified dysregulated pathways, including increased inflammatory cytokines IL-36α, IL-36γ, STAT1, and NOS2, as well as decreased levels of the immune inhibitor IL-37. Inhibition of these pathways significantly improved the lipid barrier in the HI 3D skin model, highlighting potential therapeutic targets [86].

These studies demonstrate the ability to genetically manipulate 3D skin models to investigate various conditions and phenotypes relevant to dermatological and biomedical research [81,84,85,86]. This approach offers a versatile and powerful platform to explore the mechanisms underlying cutaneous diseases, as well as to evaluate potential therapies and intervention strategies.

### 3.1. Keratinocytes, Fibroblasts, and Melanocytes

Keratinocytes, fibroblasts, and melanocytes used in the construction of skin models are often primary cells obtained from discarded surgical tissues. These tissues are largely obtained from surgical procedures such as phimosis surgery, generally in neonates [87,88], or from tissues resulting from plastic surgery (e.g., abdominoplasty), often from women [89,90]. Furthermore, some of these primary cells are commercially available in biobanks, meaning that in-house isolation is not necessary [91].

Researchers have also used models composed of immortalised cells, as exemplified in the study by Şenkal et al. (2022), where the use of the Immortalized Human Keratinocyte (HaCaT) cell line in reconstructed human epidermis (RHE) cultures demonstrated the ability to reproduce a phenotype and gene expression profile similar to those of native skin [92]. Furthermore, this model was effective for the evaluation of cell viability and chemical toxicity through skin irritation tests [92]. Mini et al. (2021) also presented consistent results when they used the HaCaT cell line to construct an RHE model [93]. Similarly, Szymański et al. (2020) obtained promising results in constructing a complete 3D human skin equivalent model using conditionally immortalised keratinocytes (Ker-CT), epidermal keratinocytes immortalised by human telomerase and CDK4, BJ-5ta human foreskin fibroblasts immortalised with human telomerase reverse transcriptase (hTERT), and melanocytes [75]. This model effectively evaluated the cytotoxicity of genipin, a collagen cross-linking agent, demonstrating non-toxicity at concentrations up to 10 μM for keratinocytes and 150 μM for fibroblasts. Additionally, the model facilitated the analysis of cytokine secretion, including GM-CSF, IL-10, IL-15, IL-1α, IL-6, IL-7, IL-8, and MCP-1, highlighting its utility in investigating skin inflammation and immune responses. These specific applications underscore the potential of this model for drug delivery studies, cytotoxicity evaluations, irritation, and sensitisation assessments, providing a valuable and ethical alternative to animal models in dermatological and toxicological research. This model demonstrated potential for use in comprehensive studies including drug delivery, cytotoxicity, irritation, and sensitisation [75].

Skin models that use stem cells have been extensively investigated, with emphasis on the use of iPSCs, as they can be obtained from a wider and more ethically acceptable source and can be reprogrammed from somatic cells. hESCs are derived from the inner cell mass (ICM) of blastocysts obtained through voluntary donations resulting from in vitro fertilisation (IVF) treatments and are strictly regulated by ethical and legal standards in several countries [94].

The differentiation of hESCs into fibroblasts and keratinocytes has been achieved through the autogenic food-free system. Fu et al. described successful differentiation into fibroblasts [95], whereas Kidwai et al. applied the same principle to obtain keratinocytes [96]. Cherbuin et al. corroborated these findings [95,96,97]. More recently, Luo et al. employed iPSC technology to differentiate them into keratinocytes (iPSC-KCs) and fibroblasts (iPSC-FBs), constructing a 3D skin model using organ-on-a-chip technology. In this model, iPSC-KCs and iPSC-FBs were cultured on a Matrigel^®^ coating, and the model presented morphology, physiology, and biochemistry similar to those of native skin [98]. The development of the skin-on-a-chip (SoC) by Luo et al. is a significant advance, standing out as a potentially applicable approach in clinical settings as well as in the cosmetic and pharmaceutical industries [98]. The SoC is a miniaturised model that combines human skin cells with microfluidic technology to simulate the structure and function of real skin, enabling a more accurate and ethical assessment of responses to chemical substances, representing a promising alternative to animal testing [98].

Skin pigmentation studies are crucial for treating pigmentary disorders. Melanocyte skin models play an important role in advancing the understanding of these phenomena. However, there are challenges in the isolation and culture of melanocytes, especially those from adult skin. Cohen et al. overcame this difficulty by isolating donor fibroblasts and reprogramming them into iPSCs, which were subsequently differentiated into melanocytes [99]. This resulted in a highly pure population of melanocytes with genomic stability and proliferative capacity. These melanocytes were used to construct a 3D skin model incorporating normal keratinocytes and fibroblasts [99]. The use of iPSC technology made it possible to overcome the challenges associated with obtaining melanocytes, resulting in an effective, reproducible model with reduced genetic variability. This model has demonstrated robust melanogenic responsiveness, providing a valuable tool for skin pigmentation studies [99]. In a subsequent study, the integration of dermal stem cell spheroids into reconstructed skin equivalents showcased the capacity of these cells to integrate into the basal layer of the epidermis and differentiate into melanocytes. These findings have prompted inquiry into whether the initial epigenetic or genetic alterations responsible for transformation occur within the dermis rather than the epidermis [100,101].

The versatility of these approaches, as demonstrated, exhibits significant potential for the development of in vitro skin models that authentically mimic the morphology and function of human skin. These models highlight the advantages and consistency of this approach in investigations that require detailed assessments of cellular phenotypes and responses in in vitro skin models. This panorama of cell procurement options, from primary cells to iPSCs, provides a robust basis for studies in diverse areas, including dermatology, toxicology, cosmetics, and pharmacology [102,103]. The variety of models available opens doors to a deeper understanding of cutaneous biological processes, which can drive significant advances in research and practical applications.

### 3.2. Dendritic Cells of the Skin

Antigen-presenting cells (APCs), including macrophages, monocytes, and dendritic cells (DCs), orchestrate immune responses in the skin; these cells reside among keratinocytes [104]. Langerhans cells (LCs) stand out as DCs of the skin that present antigens to T lymphocytes, playing a crucial role in the pathogenesis of several diseases. LCs are present in small quantities in the epidermis (2–3%) and go through different stages of maturation in the epidermis (immature), with final maturation taking place in lymph nodes (mature) [105,106,107].

Zaba et al. identified these dendritic cells by their low CD14 expression and high CD11c expression [108,109]. The activity of these cells is highly influenced by the stimulus received. For example, keratinocytes respond to inflammatory stimuli, such as interleukin (IL)-1β, lipopolysaccharide (LPS), or sensitising agents, such as chloro-2,4-dinitrobenzene (DNCB) or NiSO_4_, secreting several cytokines, such as IL-1, tumour necrosis factor alpha (TNF-α), or IL-18 [4,110,111,112]. Cutaneous DCs, including LCs and dermal DCs, are activated by this process, triggering hapten phagocytosis, followed by cell maturation and upregulation of adhesion molecules [113,114,115]. These events are crucial for cutaneous immune responses and indicate the dynamic complexity of interactions between keratinocytes and DCs in the skin.

The use of primary cutaneous DCs is challenging because of their low abundance in the epidermis and the difficulties associated with extraction and in vitro propagation, which is limited to a few passages [116]. However, some cell lines derived from human myeloid leukaemia exhibit phenotypic characteristics similar to those of DCs. It is important to note that although these cell lines possess characteristics similar to those of DCs, their responsiveness to cytokines, as mentioned previously, is generally more modest compared to that of native DCs in the skin. However, the CD34^+^ human acute myeloid leukaemia cell line (MUTZ-3) responds positively to granulocyte–macrophage colony-stimulating factor (GM-CSF), interleukin 4 (IL-4), and TNF-α. These cytokines are known to be fundamental both in vivo and in vitro for the generation of DCs from monocytes and CD34^+^ stem cells. CD34^+^ MUTZ-3 behaves like an immortalised equivalent of the CD34^+^ precursors of DCs and can acquire a phenotype similar to that of interstitial or Langerhans-type DCs. This includes the expression of a wide range of functional antigen processing and presentation pathways, especially following stimulation with specific cytokines [117,118].

Immunocompetent skin models described in the literature have focused on the incorporation of LC substitutes from the MUTZ-3 cell line [119,120,121,122]. These substitutes can originate from the CD34^+^ hematopoietic progenitor, obtained from umbilical cord blood [123], or be generated from CD14^+^ peripheral blood mononuclear cells [121,124]. This model is valuable for performing irritation tests, skin sensitisation, and assessments of allergies to substances. Its versatility allows its application in RHE models, full-thickness skin models, and SoC platforms, validating several proposed objectives [119,120,121,122].

For instance, Ouwehand et al. developed a full-thickness skin equivalent incorporating MUTZ-3-derived LCs, which, upon exposure to allergens like nickel sulphate and resorcinol, exhibited maturation and migration responses akin to native LCs, indicating the model’s utility in studying allergen-induced immune responses [120]. Similarly, Laubach et al. integrated MUTZ-3-derived LCs into a 3D full-thickness skin model, demonstrating the cells’ phenotypic plasticity and their potential in investigating cutaneous immune responses [119]. These specific applications underscore the practical relevance of MUTZ-3-based immunocompetent skin models in dermatological and toxicological research [119,120,121,122].

Another widely used cell line is the human monocytic leukaemia cell line (THP-1), which can be differentiated into immature dendritic cells (iDCs). These iDCs exhibit robust capacity and sensitivity to identify sensitising and model-classified chemicals such as DNCB and NiSO_4_ in vitro [125]. Furthermore, owing to their characteristics as APCs, these cells can phagocytose pathogen-derived membrane components and activate T cells through upregulation of IL-12p40 following sensitiser treatment [125]. Skin models using THP-1 cells have also been successfully demonstrated, highlighting their equivalence and functionality in relation to native skin in the context of sensitisation to substances [125].

Other cell lines of DCs have also been described that offer ease of cultivation and reproducible results. Among them are the human histiocytic lymphoma cell line (U-937), the human bone-derived acute myeloid leukaemia cell line (KG-1), the human promyelocytic leukaemia cell line (HL-60), and the human erythroleukaemia cell line (K562) [126,127,128]. However, these cell lines are associated with challenges such as genomic and phenotypic drift, especially after numerous passages, in addition to metabolic deficiencies and dysregulation of signalling mechanisms [126,127,128,129].

Another strategy involves the use of pluripotent stem cells, which are capable of maintaining the characteristics of primary cells along with the potential for unlimited growth and differentiation into a variety of cell types. The use of iPSCs is particularly relevant for deriving DCs, offering potential use in clinical applications and suitability for large-scale production [130,131]. However, to date, none of the epidermal or full-thickness skin models developed with iPSCs have incorporated immune cells; only 3D skin models with immortalised MUTZ-3 and THP-1 cells have been successfully constructed [119,125].

### 3.3. Endothelial Cells

Several complex skin models have been developed in recent years, and the inclusion of endothelial cells in skin models represents a significant advance, especially in simulating the in vivo vascular network [132]. The presence of these cells contributes to an authentic blood–epidermal barrier formation, resulting in models that more accurately reflect the interface between the bloodstream and the skin [132]. This allows studies on the skin permeability of chemical substances and medicines, providing valuable insights for the development of pharmaceutical and cosmetic products. Moreover, such models express immunological and inflammatory responses with greater complexity [23].

Skin microcirculation plays a fundamental role in homeostasis, thermoregulation, blood pressure, inflammatory responses, and nutrient supply. Therefore, it is crucial to incorporate vascular structures into in vitro models [133,134]. Costa-Almeida et al. performed a promising study in regenerative medicine, evaluating the ability of different types of fibroblasts to stimulate the formation of capillary-like structures when co-cultured with endothelial cells [135]. Human dermal fibroblasts (HDFs), including those from human neonatal foreskin and juvenile HDF, were co-cultured with outgrowth endothelial cells (OECs) obtained from umbilical cord blood or with human umbilical vein endothelial cells (HUVECs) [135,136]. DF, especially juvenile HDF, showed excellent efficacy in promoting the formation of capillary-like structures, regardless of the type of endothelial cells used in co-culture. Differential marker analysis indicated that specific characteristics of HDF played a significant role in this effect [135].

Another innovative approach to constructing vascularized skin models involves the use of human microvascular endothelial cells [137,138]. An equivalent vascularized skin model was developed using a vascularized biological scaffold called BioVaSc [138]. This scaffold is derived from a decellularised segment of porcine jejunum and is considered similar to a custom bioreactor system [139,140,141]. BioVaSc, implanted with human fibroblasts, keratinocytes, and human microvascular endothelial cells, presented a morphology that mimics native human skin [138]. Furthermore, it demonstrated a robust and effective barrier function, with endothelial cells lining the walls of the formed vessels, allowing perfusion with a physiological volume flow. This skin equivalent, with a vasculature similar to that seen in vivo, stands out as a sophisticated in vitro model for dermatological research, suggesting its potential for skin grafting and as an optimised alternative to animal models in pre-clinical studies [138].

Skin models using stem cells derived from human adipose tissue also represent an important platform for reconstructing hair structures in vitro [100,132]. Reconstructed tissue models, which include all three skin compartments in vivo, have been developed for efficacy testing and chemical compound mode-of-action prediction. As already mentioned, microcirculation in the skin plays a crucial role in regulating homeostasis, thermoregulation, and blood pressure, as well as directing inflammatory responses and providing nutrients and other systemic factors [142]. This vascular network, composed mainly of microvascular endothelial cells, performs important functions, such as the synthesis and secretion of chemokines and cytokines [143]. These substances activate the immune system, promoting the migration of leukocytes to inflamed areas and participating in the formation of the extracellular matrix (ECM) [144]. Therefore, the incorporation of vascular structures in in vitro models becomes crucial.

Tissue vascularisation is recognised as one of the main challenges in tissue engineering, especially in the context of reconstructed skin equivalents. This challenge has been addressed through the introduction of vascular cells, the provision of angiogenic factors, and the use of techniques such as microfluidics, bioprinting, and self-assembly [145,146,147]. However, a crucial issue related to the long-term maintenance of vascularized tissues in in vitro environments has often been overlooked. The complex interplay between growth factors and cytokines, particularly in tissues composed of more than two cell types, makes the maintenance of these tissues a challenging task. Despite this limitation, such models provide a valuable platform for studying skin diseases and testing new therapies over extended periods [23].

Although notable advances have been made, there are still challenges, such as the lack of SoC models with a simulated immune microenvironment. Current models lack LCs, macrophages, and other immune cells in the vascular system, which prevents the study of cell migration through the skin and microbiome layers, affecting permeability and other such mechanisms. Continued development of these models is essential to improving their accuracy and applicability in dermatological research [23,148].

### 3.4. Adipose Cells

Although early reconstructed skin models revolutionised animal-free testing, they were designed primarily for acute irritation and phototoxicity assays and therefore contained only epidermal keratinocytes or, at most, an epidermis-dermis construct, leaving the metabolically active adipose layer out of the equation. Over the past decade, however, evidence has shown that subcutaneous adipocytes secrete adipokines, cytokines and antimicrobial peptides that modulate inflammation, drug absorption, wound healing and barrier homeostasis. Tri-layer models that integrate vascularized adipose tissue now display gene-expression profiles, lipid metabolism and immune responses much closer to native skin, improving the prediction of chronic toxicity and transdermal pharmacokinetics. Adipose tissue cells can be acquired via abdominoplasty liposuction [149] or through mesenchymal stem cells (MSCs), which are a type of adult stem cell capable of differentiating into various cell types, such as adipocytes, osteocytes, and chondrocytes [150].

The construction of models with adipose tissue allows a comprehensive analysis of the systemic action of drugs, enabling proper study of the factors affecting the effective transfer of the compound to blood vessels and adipose tissue. The evaluation of parameters such as penetration, metabolism, and action of the test compound becomes crucial in this context [150].

The integration of adipose tissue into these models not only enhances the understanding of the action of specific compounds but also promotes advances in the search for effective therapeutic solutions [100,150].

Jäger et al. developed a complete 3D skin model with three layers. The model presented morphology and physiology similar to those of native skin in vivo, with dermal and epidermal layers, as well as the synthesis of leptin, a marker of adipose metabolism. This model with three layers was developed to overcome fibroblast-mediated contraction, which is one of the challenges encountered in 3D models that incorporate both dermis and epidermis [151].

Additional studies involving adipose tissue-derived stem cells (ADSCs) observed that these cells had the capacity for angiogenesis, migration, differentiation into fibroblasts, and positive regulation of macrophage chemotaxis, processes that are extremely important for healing. Furthermore, the production of growth factors associated with these processes has been noted in several studies [152,153,154,155]. Additionally, the paracrine function of ADSCs in promoting angiogenesis and generating growth factors that stimulate neovascularisation has been reported [153,156,157,158], and the combined release of vascular growth factors and fibroblast growth factors favours dermal fibroblast reconstitution, promoting wound healing [159,160].

### 3.5. Cutaneous Appendages

Cutaneous appendages, such as sweat glands (SGs) and hair follicles (HFs), play essential roles in maintaining skin homeostasis, including thermal regulation, immune defense, secretion of bioactive factors, and support for tissue regeneration. The interaction between these appendages has been directly implicated in wound healing and cutaneous re-epithelialization but is still incompletely understood due to the scarcity of in vitro models that comprehensively recapitulate their morphogenesis and function [133]. Recent advances in tissue engineering have enabled the development of more complex 3D skin models, which not only include the multilayered epidermis and a living dermis rich in fibroblasts and extracellular matrix but also incorporate functional appendages, such as HFs and SGs. This incorporation significantly expands the applicability of these models in toxicological, pharmacological, cosmetic, and regenerative medicine testing [161].

In the case of hair follicles, in vitro reconstruction involves cocultures of primary keratinocytes or induced pluripotent stem cell (hiPSC)-derived cells with dermal papilla cells or crest-like neural cranial mesenchyme. This arrangement reconstitutes the essential structures of the HF, including the bulb (ALP^+^/SOX2^+^), proliferative matrix (Ki-67^+^), and epithelial sheaths, and can produce keratinised shafts that enter a growth cycle after grafting [162,163]. More recent work, such as that by Motter et al., has demonstrated the feasibility of bioprinting follicular units using cellular spheroids, establishing a three-dimensional architecture similar to human skin with integrated functional HFs, which represents a significant advance towards more realistic regenerative models [164].

Complementarily, eccrine SGs have been regenerated from specific precursor lines, such as the EC23 line, or derived from hiPSCs, forming duct-secretory spirals that express markers such as CK8/CK18, AQP5, and CFTR, in addition to myoepithelial α-SMA^+^ cells. These structures are capable of secreting functional sweat in response to cholinergic stimuli, demonstrating their functionality in vitro [165,166]. The integration of these glands into cutaneous models represents an important milestone in the development of more physiologically relevant equivalents, since sweating is directly linked to thermoregulation, cutaneous defence, and the cellular microenvironment.

Recent studies have focused on the interaction between these two appendages. Zhang et al. (2021) [133] for example, developed a model combining bioprinted scaffolds containing mesenchymal stem cells and dermal homogenate with trichogenic spheroids of keratinocytes and fibroblasts, obtaining a tissue that simultaneously reproduced functional HFs and SGs. After 7 to 14 days of culture, the construct expressed follicular (KRT17, ALP, and CDH3) and glandular (KRT18/19, AQP5, and α-SMA) markers, demonstrating the maturation of both appendages. Functional analyses showed that HFs positively influenced SG differentiation, while the glandular microenvironment primarily favoured sweat maturation. These findings highlight the crucial role of specific epithelial–mesenchymal interactions and the microenvironment in the coordinated formation of cutaneous appendages [133].

Thus, the integrated and functional presence of HFs and SGs within cutaneous models with organised epidermis and dermis represents the contemporary criterion for defining an “advanced skin model.” Such models offer versatile platforms to simultaneously study barrier, secretion, wound healing, immune response, and thermoregulatory physiology, expanding their translational potential for clinical and industrial applications [161,167].

## 4. Applications of In Vitro Skin Models

Several laboratories around the world have attempted to construct in vitro skin models, targeting various applications [161,168]. These models include several types such as RHE, skin equivalent, single-cell, or multi-cell models, as well as models that simulate both healthy conditions and specific dermatological diseases (e.g., Atopic dermatitis, psoriasis) [169]. These structures can be constructed manually or using advanced techniques, such as bioprinting, and can be implemented in different formats, such as culture plates or microfluidics. Each model is designed with a specific purpose, highlighting the diversity of approaches and applications to meet varying research and development needs (Table 2).

Three-dimensional culture models are complex systems that mimic the morphological and functional characteristics of in vivo tissues and can be monotypic (models that use a single cell type) or multitypic (co-culture of different cells) [170]. In vitro RHE is a 3D culture model that mimics the human epidermis and can vary in its cellular composition. Most RHE models found in the literature are monotypic models built exclusively with keratinocytes [89,90,171,172,173,174]; however, there are also multitypic models (co-culture with melanocytes for pigmented models or with LCs for immunological models) [175,176,177,178].

Although the thickness of the skin in vitro (in micrometres and/or by number of layers) is informative for comparing models with human skin, these data are reported heterogeneously. In commercial RHEs (EpiSkin, SkinEthic, EpiDerm), the literature tends to emphasise histological similarity and the presence of a functional stratum corneum, but without standardising numerical thickness values, as these vary from batch to batch [179]. In full-thickness skin models, some articles describe layers/architecture with scale bars (for example, T-Skin™ reports stratified epidermis with ~8–12 living layers under the stratum corneum, with 100 µm bar figures), but do not always provide the average thickness in µm of the entire tissue [180]. In skin-on-a-chip, studies highlight improved morphogenesis and barrier function under perfusion, but often without quantifying the absolute thickness of the epidermis/dermis in the main text [181]. In 3D bioprinting, there are detailed descriptions of architecture (e.g., inclusion of rete ridges), but thickness values vary depending to printing parameters and construct design, reinforcing the need to report protocol and scale along with the values when available [182].

**Table 2 pharmaceutics-17-01342-t002:** Comparison of in vitro human skin models: cell type employed, structural classification, validation status (OECD) and application.

In Vitro Skin Model	Cell Types	Skin Model Type	Model Validated by the OECD	Model Application
KeraSkin™ [176,183,184,185,186]	Primary human keratinocytes	Reconstructed epidermis	Yes	Irritation test (TG 439), phototoxicity, Genotoxicity
Reconstructed Human Epidermis [187]	Primary human keratinocytes	Reconstructed epidermis	No	Skin barrier function and hydration, Skin irritation, Skin corrosion, UV exposure, DNA damage, Bacterial adhesion, Omics, Permeability, and disease studies
SkinEthic™ RHE [174,188,189,190,191,192]	Primary human keratinocytes	Reconstructed epidermis	Yes	Skin irritation, Skin corrosion, Medical devices, UV Exposure, DNA Damage, Bacterial adhesion, Omics, Permeability,
EpiSkin™ Human Epidermis [175,179,193,194,195,196]	Primary human keratinocytes	Reconstructed epidermis	Yes	Skin irritation, Skin corrosion, UV Exposure, DNA Damage, Bacterial adhesion, Omics, Permeability
LabCyte EPI-MODEL [197,198,199,200,201]	Primary human keratinocytes	Reconstructed epidermis	Yes	Skin irritation, Skin corrosion, Skin sensitisation, Molecular analyses, Evaluation of biological processes
EpiDerm™ [202,203,204,205]	Primary human keratinocytes	Reconstructed epidermis	Yes	Skin irritation, skin corrosion, Skin Hydration, Dermal Drug Delivery, Phototoxicity, Dermal Genotoxicity, Epidermal Differentiation
Straticell RHE [206,207]	Primary human keratinocytes	Reconstructed epidermis	No	Safety and efficacy testing of topical products, skin penetration studies, and interactions with skin microorganisms
In house reconstructed human epidermis (ALI) [88,89,208,209,210]	Primary human keratinocytes	Reconstructed epidermis	No	Analyses of toxicological studies, assessment of skin phototoxicity, biochemical studies, and disease studies
Straticell RHE-MEL [211]	Primary human keratinocytes and melanocytes	Reconstructed epidermis (pigmented)	No	Evaluation of depigmenting ingredients and investigation of pigmentation disorders
KeraSkin-M™ [176,177,178,183,184,212]	Primary human keratinocytes and melanocytes	Reconstructed epidermis (pigmented)	No	Irritation test (TG 439), Regulation of skin pigmentation
SkinEthic™ RHPE [213,214,215,216]	Primary human keratinocytes and melanocytes	Reconstructed epidermis (pigmented)	No	Pigmentation, Depigmentation, Omics, UV Exposure
MelanoDerm™ [217,218,219,220]	Primary human keratinocytes and melanocytes	Reconstructed epidermis (pigmented)	No	Skin Brightening, Pigmentation Studies
SkinEthic™ RHE-LC [120,216,221,222]	Primary human keratinocytes and Langerhans cells	Reconstructed epidermis & Langerhans	No	Skin Immune Response, UV Exposure, Bacterial adhesion, Omics, Permeability
Straticell RHE-huSN [223]	Primary human keratinocytes and iPSC-derived sensory neurons	Reconstructed epidermis & sensory neurons	No	Neurocutaneous interactions, assessment of soothing properties of dermocosmetic ingredients, and investigation of sensory mechanisms
KeraSkin-FT™ [183,224,225]	Primary human keratinocytes and fibroblasts	Full thickness skin (epidermis & dermis)	No	Irritation test (TG 439), Microbiome, ageing
Full Thickness Skin Model [226]	Primary human keratinocytes and fibroblasts	Full thickness reconstructed skin tissue (epidermis & dermis)	No	Skin irritation, skin sensitisation, genotoxicity, Pigmentation, skin barrier function and moisturising, anti-ageing, stress/inflammation, UV protection
T-Skin™ [180,188,227,228,229]	Primary human keratinocytes and fibroblasts	Full thickness reconstructed skin tissue (epidermis & dermis)	No	UV Exposure, DNA Damage, Bacterial adhesion, Omics, Permeability
EpiDerm™ FT [230,231,232,233]	Primary human keratinocytes and fibroblasts	Full thickness reconstructed skin tissue (epidermis & dermis)	No	Anti-ageing, Wound Healing, Skin Hydration, UV Protection
Phenion^®^ FT Skin Model [234,235,236]	Primary human keratinocytes and fibroblasts	Full thickness reconstructed skin tissue (epidermis & dermis)	No	Used for basic and clinical dermatological research, cosmetic claim support, drug delivery and penetration studies, wound healing, toxicity testing, and evaluating environmental effects on skin physiology
Phenion^®^ FT AGED [234,237,238]	Primary human keratinocytes and fibroblasts	Full thickness reconstructed skin tissue (epidermis & dermis)—senescent	No	Skin ageing and testing the efficacy of anti-aging products.
Phenion^®^ FT LONG-LIFE [234,239]	Primary human keratinocytes and fibroblasts	Full thickness reconstructed skin tissue (epidermis & dermis)—aged	No	long-term studies, enabling evaluation of delayed effects, repeated substance exposure, skin regeneration, and in vitro research on skin tumour development and treatment.
Skimune 3D^®^ [240,241]	Primary human keratinocytes and fibroblasts	Full thickness reconstructed skin tissue (epidermis & dermis)	No	Assessment of toxicity and safety of drugs, cosmetics, chemicals, and biologics. It evaluates immune responses, including dose toxicity, cell viability, apoptosis, necrosis, and effects on cytokines, growth factors, and chemokines, helping to identify immune reactions missed by simpler assays.
Pigmented Skin Model [99,242,243,244]	Primary human keratinocytes, fibroblasts and melanocytes	Full thickness reconstructed skin tissue (epidermis & dermis, pigmented)	No	Skin pigmentation, Pigmentation disorder, Skin ageing
Neurodermatology Cell Model [245]	Primary human keratinocytes, sensory neurons	Full thickness reconstructed skin tissue (epidermis & dermis) incl. sensory neurons	No	Skin ageing, Inflammatory skin disorders, Soothing, Skin pigmentation Photo-aging, Warming sensation, Microcirculation, Hair growth
Vascularised Human Skin Equivalent [224]	Primary human keratinocytes, dermal fibroblasts, HUVECs, preadipocytes, mature adipocytes	Bioprinting vascularised human skin equivalent	No	Development of cosmetics, topical treatments for skin rejuvenation, and therapies to improve skin health and appearance in ageing populations
In house full-thickness skin [91]	Primary human keratinocytes, diabetic dermal fibroblasts, diabetic adipocytes	Full-thickness skin with vascularised hypodermis	No	Diabetic skin model, Insulin resistance, Inflammation, Skin disease modelling, Drug testing
Vascularised Human Skin Equivalent [246]	Primary human keratinocytes, dermal fibroblasts, cord blood-derived endothelial cells, placental pericytes	Bioprinting vascularised human skin equivalent	No	Regenerative medicine (skin grafts)
Skimune^®^ [241,247]	Full thickness skin tissue (explant) incl. Peripheral Blood Mononuclear Cells	Full thickness reconstructed skin tissue (explant)—dermis, epidermis; incl. Blood samples	No	Assessment of immunotoxicity and hypersensitivity in response to chemicals, drugs, biologics, and cell or gene therapies, using autologous skin and blood to predict human immune responses.
Ex vivo explants [248]	Full thickness skin tissue (explant)	Full thickness skin tissue (explant)—dermis, epidermis	No	Anti Aging Pigmentation Stress/Inflammation UV Protection
HypoSkin^®^ [249,250]	Full thickness skin tissue (explant)	Full thickness skin tissue (explant)—dermis, epidermis & hypodermis (adipose)	No	Evaluation of local injection reactions, testing injectable formulations’ safety, investigating immune mechanisms, and conducting molecular analyses to understand skin responses
NativeSkin^®^ [251,252,253,254]	Full thickness skin tissue (explant)	Full thickness skin tissue (explant)—dermis, epidermis	No	Cosmetic claims, Skin delivery, Toxicity, Wound healing, Skin microbiome
Full-thickness skin equivalent with integrated hair follicle [164]	Primary human keratinocytes, dermal fibroblasts, dermal papilla cells (DPCs), human umbilical vein endothelial cells (HUVECs), melanocytes	Full-thickness skin equivalent with integrated hair follicle-like structures via 3D bioprinting	No	Hair follicle formation, Skin appendage modelling, Drug testing, Cosmetic testing, Regenerative medicine
Full-thickness human skin-on-a-chip with enhanced epidermal morphogenesis and barrier function [181]	Primary human fibroblasts and N/TERT human keratinocytes	Full-thickness human skin equivalent on chip, with fibrin-based dermal matrix and dynamic perfusion.	No	In vitro tool for basic research and pharmaceutical, toxicological, and cosmetic sectors

Although keratinocytes secrete the basal lamina and remain anchored to it, the absence of cellular interaction with fibroblasts hinders the construction of an RHE. Fibroblasts significantly influence epidermal structuring in vivo by promoting keratinocyte differentiation [255]. Therefore, constructing an RHE in vitro requires the use of a culture medium supplemented with various additives that promote keratinocyte proliferation and differentiation, in combination with traditional culture systems and an air–liquid interface. This condition facilitates cellular differentiation, enabling the development of an epidermal model containing all four in vivo epidermal layers: the basal layer, spinous layer, granular layer, and stratum corneum [255]. Such an in vitro RHE represents the function of the epidermis more accurately than 2D keratinocyte cultures, as it allows for the topical application of chemical substances to the outermost layer of the skin, the stratum corneum [132,256,257].

The RHE replicates the structure and complexity of the in vivo epidermis and can also demonstrate several measurable epidermal functions, such as water loss and transepithelial electrical resistance (TEER). Therefore, this model can be used for various studies, mainly those analysing dermal exposure, such as the evaluation of irritants [88,208,209,258,259] and skin sensitisers [260]. Moreover, it can also be used for metabolic analyses of products [210], analyses of cutaneous immunotoxicological responses [209] and toxicological studies [174], assessments of skin phototoxicity [261], and biochemical [210] and disease studies [262].

To date, seven validated RHEs have been recognised by the OECD and are widely marketed in Europe, North America, and Asia. These are EpiDerm^®^ (MatTek Corporation) [21,204], EpiSkin^®^ (L’OREAL) [21,175], SkinEthic^®^ (L’OREAL) [21,191], LabCyte EPI-MODEL (Japan Tissue Engineering Co., Ltd., Aichi, Japan) [21,198], epiCS^®^ (CellSystems^®^) [21,263], Skin+^®^ (Sterlab) [21], and KeraSkinTM SIT (BioSolution) [21,176,183,184,212].

Further, there are more elaborate skin models, such as human skin equivalents (HSE), in which different cell types, such as keratinocytes and fibroblasts, are incorporated to investigate additional functions, especially for dermatological medicines, cosmetics, and pharmaceuticals [256]. These models include other cells such as melanocytes, responsible for skin pigmentation [99,243,244], LCs, which are epithelial DCs fundamental to the dermal immune system [245,264], endothelial cells, which contribute to the formation of blood capillaries [224,246], adipose cells [91], and hair follicles [164].

### Pharmaceutical Applications

For topical and transdermal products, IVPT remains a cornerstone to quantify drug delivery from semisolids and patches and to compare formulations under controlled conditions. Robust study design (diffusion-cell geometry, receptor-phase composition, sampling/analytics) and skin integrity testing (TEWL, electrical resistance, or ^3^H_2_O) are critical to generate meaningful performance data and support rational IVIVC. The TEWL varies widely with anatomical site, instrument, and environment; therefore, there is no single “standard” value for intact human skin, although practical thresholds in IVPT typically adopt cutoffs of up to approximately 15 g m^−2^ h^−1^, depending on the method. In RHE models, TEWL is useful as a barrier indicator, but values depend on the model and protocol, always requiring reporting of the device and measurement conditions [265,266,267]. Recent best-practice papers emphasise pre-defined integrity criteria, careful handling of flux/lag-time calculations, and transparent reporting to improve cross-lab reproducibility [265,268,269].

In vitro–in vivo correlation (IVIVC) in topical products aims to demonstrate that metrics obtained in in vitro assays predict performance in the body, allowing for the replacement or reduction in clinical equivalence studies when scientifically justifiable. For cutaneous products, IVIVC typically integrates in vitro release (IVRT) and in vitro permeation through human skin (IVPT) data with local in vivo readings such as tape stripping, TS, drug-specific pharmacodynamics, and, increasingly, physiology-based pharmacokinetic models (PBPK) to translate fluxes and quantities measured in ex vivo skin into local and systemic profiles in humans. The actual regulatory guidelines for local cutaneous products in Europe formalise the role of IVRT and IVPT in demonstrating equivalence, offering a structured pathway for constructing IVIVC when critical quality characteristics and experimental design are rigorously controlled [24]

The choices of skin, diffusion cell, dose, and receptor conditions impact the predictive capacity of IVRT [265,268]. Furthermore, some authors demonstrate that using dermal PBPK to link IVRT and IVRT to human outcomes reduces uncertainties by incorporating vehicle and active ingredient properties, partitioning, and diffusion through skin layers. These demonstrates that PBPK are capable of translating in vitro data to in vivo scenarios with minimal calibration and better predictive reliability [270,271]. Another important point to consider is anchoring the correlation in sensitive in vivo endpoints, for example, TS in volunteers, which has already been shown to be associated with IVRT in metronidazole creams, serving as an experimental bridge between the bench and the clinic [272]. Together, these elements strengthen IVIVC for topics and are aligned with the regulatory movement that prioritises integrated batteries of new approach methods in cutaneous products [24].

Microphysiological (“skin-on-a-chip”) models add perfusion and shear to conventional 3D skin equivalents, enabling dynamic exposure scenarios, co-culture with endothelial/immune components, and longer-term readouts of barrier, permeation and inflammation. State-of-the-art reviews in Pharmaceutics (2022), Lab on a Chip (2023) and JID (2024) outline how SoC may de-risk formulation decisions and provide animal-free dermal absorption platforms complementary to human/porcine skin and RHE/HSE models [23,273,274].

Skin-on-chip validation criteria depend on the context of use and target metrics with acceptance criteria, as recommended in microphysiology standardisation initiatives and regulatory workshops [275,276]. Device and perfusion conditions must be verified, as well as barrier confirmation using complementary indicators such as real-time for electrical integrity, permeability assays with hydrophilic or lipophilic tracers, histology, and immunostaining of epidermal differentiation.

Transepithelial electrical resistance (TEER) of excised human skin used in IVPT typically adopts integrity thresholds of ≥10 kΩ cm^2^, but the FDA itself emphasises that absolute values depend on the electrodes, frequency, and effective area, recommending method-dependent criteria. For other species, comparative studies indicate integrity references of approximately 5 kΩ cm^2^ for mice and guinea pigs, 4 kΩ cm^2^ for swine, 3 kΩ cm^2^ for rats, and 0.8 kΩ cm^2^ for rabbits [277]. In contrast, intact human skin in vivo can exceed 50 kΩ cm^2^. In reconstructed models, mature RHE values typically range from 1 to 3 kΩ cm^2^, with averages reported around 1.6 to 1.7 kΩ cm^2^ for models comparable to those validated by the OECD, while full-thickness equivalents tend to have lower values, around 0.7 to 0.9 kΩ cm^2^. When RHE is performed with immortalised lines, values can drop substantially; for example, KerCT may not reach ranges considered adequate even when presenting claudins. Interlaboratory heterogeneity reinforces the need to standardise protocols and report measurement conditions, including electrode type, frequency, temperature, and area [181,278,279].

These three axes TEER, permeability, and labelling are recognised as central tools for assessing barriers in organ-on-chip systems [280,281]. For calibration and quality control, reference compounds and formulations, including classic markers such as tritiated water, are used, and permeability databases help quantify intra- and interindividual variability and establish acceptance ranges prior to comparative studies [265]. And when the goal is clinical predictability, evidence is integrated with physiologically-based pharmacokinetic models (PBPK) to translate in vitro measured fluxes and quantities into human local or systemic profiles [270,282].

Despite the widespread use of TEER as a readout of integrity, numerically matching RHE or RHS to excised human skin is challenging for methodological and biological reasons. The absolute value of TEER depends on experimental factors such as temperature, geometry and effective membrane area, electrode positioning, and measurement frequency, which introduces significant heterogeneity between laboratories and hinders direct comparisons with native skin [283,284,285].

Furthermore, structural and compositional differences between reconstructed models and ex vivo skin affect electrical conduction and response to perturbations, so that functional barrier equivalence does not always translate into TEER equivalence [280,286].

From a regulatory perspective and in toxicological and cosmetic practice, internationally accepted RHE irritation methods base their acceptance on performance standards and primary functional readings. Barrier and irritation assessment in reconstructed epidermal models typically uses viability by MTT and time to toxicity (ET50) with controls such as Triton X as primary endpoints. X-100, often complemented by histology and differentiation immunostains. This framework is reflected in the guidelines and performance standards associated with OECD TG 439 and in widely used validated protocols rather than in universal TEER thresholds, which reinforces the use of TEER as a complementary and context-dependent metric [21].

The regulatory landscape increasingly references these in vitro approaches: the FDA (2022) [35] provides principles for IVPT in ANDAs, while the EMA (2024) [24] details quality/equivalence expectations for locally applied, locally acting cutaneous products (including IVRT/IVPT, tape-stripping and cutaneous PK) [24,26]. In parallel, OECD TG 439 and performance-standard updates strengthen the acceptance of RHE-based irritation models within integrated strategies, reinforcing the role of in vitro systems as NAMs in pharmaceutical development.

## 5. In Vitro Models to Study Skin Diseases

In vitro skin models have become powerful platforms for investigating various dermatological conditions, including inflammatory disorders such as psoriasis, atopic dermatitis (AD), and vitiligo [287], various forms of skin damage, such as wounds, scars, burns, and photodamage [256], as well as neoplastic transformations such as melanoma [283]. These models offer a physiologically relevant simulation of human skin pathologies, supporting more specific, ethical, and reproducible approaches in dermatological, pharmaceutical, and toxicological research [288].

A comprehensive review by Sarama et al. (2022) [289] highlights how both psoriasis (PS) and atopic dermatitis (AD) have been effectively modelled using different in vitro strategies, including 2D cocultures, full-thickness human skin equivalents (HSEs), and advanced skin-on-a-chip microphysiology platforms. These systems successfully recapitulate hallmark pathological features such as keratinocyte hyperproliferation, parakeratosis, Th17/Th2-mediated inflammation, barrier dysfunction, and immune cell infiltration. Furthermore, they allow detailed histological evaluation, molecular profiling of gene and protein expression (e.g., IL-17A, S100A7, and filaggrin), and functional permeability assays, providing a robust translational alternative to animal testing [289].

Furthermore, Jang et al. (2023) reviewed recent progress in 3D in vitro models for AD that capture key features of the disease—such as a Th2-skewed cytokine environment, reduced filaggrin expression, T-cell infiltration, and dysbiosis—in platforms such as skin organoids and skin-on-a-chip microfluidic systems [290]. These complex models enable mechanistic studies and preclinical screening of topical therapies, employing histological staining, ELISA, immunofluorescence, and transcriptomic analyses to assess the efficacy and mechanistic pathways of candidate treatments.

Together, these studies reinforce the growing relevance of in vitro skin disease models as alternatives to in vivo experimentation, offering accurate and human-relevant systems for studying disease mechanisms and testing therapeutic strategies.

## 6. Advances in Skin Models

As the limitations of extrapolating data from animal models to humans owing to phylogenetic differences are increasingly recognised, the demand for alternative testing methods has grown among researchers [23,291,292,293,294,295]. However, aspects such as reliability, reproducibility, and translational potential must be considered in the development of alternative models. Therefore, new methodologies were developed that aimed to achieve a high similarity level between the model and the native organ, in addition to addressing issues such as scalability and controlled manipulation. In this context, a wide variety of techniques have been developed to cultivate cells in 3D structures and have been excellently described in several articles [23,291,292,293,295]. These techniques include bioprinting and microfluidics platforms, which have been useful tools to reduce the dependence on animal models in research. These promising approaches offer significant advantages, allowing the simulation of physiological conditions that bear greater resemblance to the human environment and facilitating the performance of more accurate and clinically relevant studies. Furthermore, the use of these technologies can contribute to a significant reduction in the use of animals in experiments, aligning with ethical principles and promoting faster advances in research in different areas [183].

Three-dimensional (3D) printing has significantly impacted diverse fields such as engineering, manufacturing, and medicine, including the advancement of alternative pre-clinical testing methods. Within this context, the development of biocompatible 3D printing systems has shown promise for tissue engineering applications. Specifically, 3D bioprinting, a technique involving the precise layer-by-layer deposition of bioinks comprising living cells, extracellular matrix analogues, and bioactive molecules, stands out as a powerful tool in tissue engineering. It allows cells to grow and mature within biocompatible, controlled environments, forming functional tissues that accurately recapitulate the complex spatial relationships present in native biological structures [292].

This technology has revolutionised the production of skin models, making it possible to scale up and automate their fabrication [296,297]. Compared to traditional, manually produced skin equivalents, 3D bioprinted models offer benefits, as to employ chemically defined bioinks. These chemically defined materials eliminate variability typically associated with animal-derived components and substantially reduce the risk of pathogen transmission, enhancing the overall safety and consistency of the models [298].

Despite technological advances, several challenges in 3D bioprinting still need to be addressed. Among these are the complex preparatory conditions required for bioprinted skin constructs and the difficulties associated with achieving sufficient stability and robustness of biofabricated skin, which currently limit wider application and consistent results [292].

Another advancement is the culturing of cells on microfluidic chips, often referred to as ‘organs on a chip’ (OoC), which represent dynamic models that offer a controllable environment. The SoC technologies, which are nothing more than a skin built on a microfluidic system, can control parameters such as topography, fluidic shear stress, and culture perfusion [299]. These scaffold-free microfluidic models significantly improve the mimicry of organs present in living organisms, as they receive a continuous supply of nutrients and oxygen while also removing metabolic waste, creating an artificial network. This approach enables the production of bulky tissues and the assembly of multiple organoids or spheroids, allowing the in vitro creation of complete systems [300].

In addition to conventional OoC, 4D skin models are emerging, constructions that incorporate stimulus-responsive biomaterials and bioinks (e.g., shape-memory polymers and programmable hydrogels) capable of changing shape, properties, or function over time under the influence of light, temperature, pH, enzymes, or external fields. These systems have been explored for wound healing, infection control, and local factor delivery, as well as for dynamic barrier models when integrated with microfluidics (4D SoC). However, their commercial availability is still in its infancy compared to RHEs and traditional full-thickness models, most examples remain on research platforms, facing challenges of standardisation, durability, long-term biocompatibility, and regulatory validation. The Literature consolidates the concept and repertoire of smart materials for 4D skin, while work with full-thickness SoC demonstrates how perfusion and precise control of the microenvironment improve epidermal morphogenesis and barrier function, characteristics that 4D constructs seek to make adaptable and programmable [301,302,303].

The advancement of SoC technology has enabled the transition from static 3D cultures to dynamic models that more closely resemble human physiology [23]. Among the advantages of the microfluidics system, its ability to provide precise culture conditions and improved cell monitoring stands out. Furthermore, this system requires a few cells for cultivation, which minimises reagent consumption. The ability to maintain highly controlled conditions, along with the automation capability, significantly reduces the time needed to perform experiments, ensuring high throughput and the ability to perform real-time analyses. However, it is important to highlight some disadvantages, such as the challenges faced in analytical chemistry and the high costs associated with maintaining and running the system [23,304].

There is a continuous need for innovation and improvement in the area of 3D skin models, aiming to overcome challenges and maximise their potential in several areas, including tissue engineering, biomedicine, toxicology, and the cosmetic and pharmaceutical industries. A report by Grand View Research, published in 2016, analysed the global 3D bioprinting crop market, estimating its value at US$558.0 million in 2016 with a compound annual growth rate (CAGR) of 14.8%, predicting that it would reach US$2.6 billion in 2024. However, more recent reports indicate a much more significant growth than initially estimated [304]. In 2023, the 3D bioprinting market was valued at US$20.37 billion, with a CAGR of 23.3% predicted for the 2023–2030 period (Grand View Research, 2022) [305]. These data reflect the growing demand for prototyping applications in various industry sectors, with an emphasis on the healthcare sector.

Despite representing a noteworthy advance in science, in vitro skin model constructs still present significant challenges that need to be overcome. Although these models enable cellular interactions and mimic some properties of human skin, such as its permeability, they still cannot adequately reproduce the natural dynamics of skin. Other important limitations include the lack of a functional immune system and the absence of blood vessels, crucial elements for maintaining adequate physiological conditions. However, the incorporation of vascularisation may offer a promising solution to expand the viability of these skin models. It is also important to highlight that these models (e.g., 3D bioprinting and SoC) have higher costs than manual methods [306,307].

## 7. Challenges and Conclusions

Three-dimensional skin models have vast potential to accurately reproduce the morphology, physiology, and biochemistry of human skin and are essential for in vitro investigation of cellular mechanisms, safety assessments, studies on skin diseases, and drug testing. With the evolution of technologies such as bioprinting and OoC, these models have become even more complex and comprehensive, incorporating multiple cell types in dynamic and variable environments. These advances make it possible to observe the interaction between cells, more faithfully replicating the natural events that occur in human skin [308].

Current in vitro methods have different degrees of regulatory maturity, which directly influences their experimental selection. RHE models are the most established for regulatory purposes, with specific OECD guidelines for skin irritation (TG 439) and corrosion (TG 431), widely accepted for hazard classification according to the GHS, which promotes reproducibility, interlaboratory comparability, and routine use in the chemical and cosmetic industries [21,309].

Full-thickness skin models expand barrier physiology and allow for additional purposes, such as genotoxicity in the 3D Skin Comet with Phenion FT, already followed by EURL ECVAM, but still with a more restricted validation scope than RHE and, therefore, presenting less trans-regulatory standardisation [310].

Currently, approaches to skin sensitization do not have a single, individually validated 3D model. Regulatory acceptance falls to Defined Approaches and IATA, which combine in chemico and in vitro methods according to APO for sensitization, according to OECD GD 497 and related documents, providing robust decision-making but requiring integration of multiple sources and expertise for interpretation [311,312,313].

Systems-on-chip (SoC) and bioprinting offer dynamic microenvironments, perfusion, and architectural control, with the potential to address vascularization and immunocompetence gaps. However, they still lack harmonised protocols, multi-site evidence, and specific OECD guidelines, limiting their immediate regulatory use. Dermal absorption assays in diffusion cells are covered by OECD TG 428, demonstrating the acceptance of in vitro methodologies for transdermal kinetics, although extrapolation to advanced 3D models still requires further standardisation [314].

Indeed, RHE is the most validated option and the one with the most regulatory readiness in terms of irritation and corrosion, full-thickness expansion mechanisms, and endpoints but with less standardisation, while SoC and bioprinting are advancing in physiological relevance, although they are still in the formal validation phase.

In order to accelerate the adoption and standardisation of advanced skin models, researchers should follow good in vitro method practices (GIVIMP) for the design, quality control, and detailed reporting of critical variables such as TEER, thickness, and viability, favouring reproducibility and interlaboratory studies. Industry should prioritise RHE in already validated scenarios for irritation and corrosion, as per OECD TG 439 and OECD TG 431, and integrate dermal absorption by diffusion cells, as per OECD TG 428, when assessing exposure. Regulators should expand guidelines for Defined Approaches in sensitization (OECD GD 497) and promote specific roadmaps for SoC and bioprinting aligned with GIVIMP, accelerating the transition from proof of concept to regulatory acceptance [315].

Despite notable advances in the development of in vitro skin models, challenges remain in achieving complete equivalence with in vivo models, especially in capturing complex systemic responses. Additionally, we face the significant challenge of seeking alternative methods that eliminate the need for animals and animal-derived reagents in skin models and other cultured organs [316]. These reagents, used in several stages, from supplementation of the culture medium to the outcome of the assay, include components of animal origin, such as serum, Matrigel™, ECM proteins, and antibodies, among others. The use of these products presents notable challenges, such as immunogenic potential and biological variability, owing to their animal origin, compromising the replicability and validity of the results, in addition to raising ethical considerations related to animal welfare. Furthermore, these reagents may contain pathogens and exhibit batch-to-batch variability, impacting the reproducibility of the data obtained [295].

Ethical considerations regarding the use of these reagents reflect the growing concern for animal welfare, driving the continued need to develop more ethical and sustainable alternatives for the cultivation of in vitro models. These challenges highlight the crucial importance of exploring alternatives free of animal-derived components, aiming to improve the reliability, consistency, and integrity of the in vitro models developed [295].

In this context, in silico methods play an important role in the analysis of chemical substances, candidate medicines, and biological interactions, significantly contributing to reducing the use of in vitro models and, consequently, the dependence on animal-derived products. Although in vitro and in vivo assays are still required to generate and validate in silico data, this approach enables time and cost savings by facilitating the development of chemically defined media [295].

Several chemical substitutes for animal-derived reagents have been studied, with some showing promising results, such as the protein sericin, a byproduct of the silk industry, as a substitute for foetal bovine serum (FBS), and recombinant antibodies that can be produced without an animal host [317,318]. However, it is imperative to conduct more studies on these alternative methodologies with robust validations to gain acceptance among researchers and the scientific community at large.

Although there are effective methods that meet specific objectives, especially in the area of risk and safety assessment, it is still common to observe a gap between scientific advances and the updating of legislation in several countries. In addition, many researchers are still resistant to adopting animal-free approaches, which constitutes a major challenge for the consolidation of ethical and innovative scientific practices. Overcoming this barrier requires not only greater awareness but also the generation of robust scientific evidence that strengthens the confidence of professionals in the field and supports the formulation of public and regulatory policies favourable to the use of alternative methods [316].

Policy momentum now favours NAMs in drug development. The FDA Modernization Act 2.0 (2022) authorises non-animal alternatives for IND/BLA submissions, and the FDA’s 2025 Roadmap charts pathways to reduce animal use in preclinical safety. For skin models, the near-term priorities are (i) standardised IVPT/IVRT protocols with shared integrity thresholds and reporting templates; (ii) benchmarking HSE/SoC barrier and metabolism versus native skin; (iii) integrating immune/vascular components under microfluidic perfusion to capture disease-relevant endpoints. Together with evolving EMA/FDA guidance, these advances should accelerate adoption of in vitro skin NAMs for topical bioequivalence, risk/safety and efficacy screening [20,24,245].

In this context, it is crucial to invest in developing more advanced and representative experimental models capable of simulating the skin both morphologically and functionally, while incorporating elements of other systems such as the nervous, immune, and circulatory systems. This complexity is vital for achieving functional equivalence that can build enough confidence to significantly reduce, or even replace, the use of animal models. In addition, it is crucial to improve in vitro cell durability, allowing long-term tests to be performed, such as chronic or repeated-dose tests. With these advances, it will be possible to promote a more modern, precise science free from animal experimentation.

## Figures and Tables

**Figure 1 pharmaceutics-17-01342-f001:**
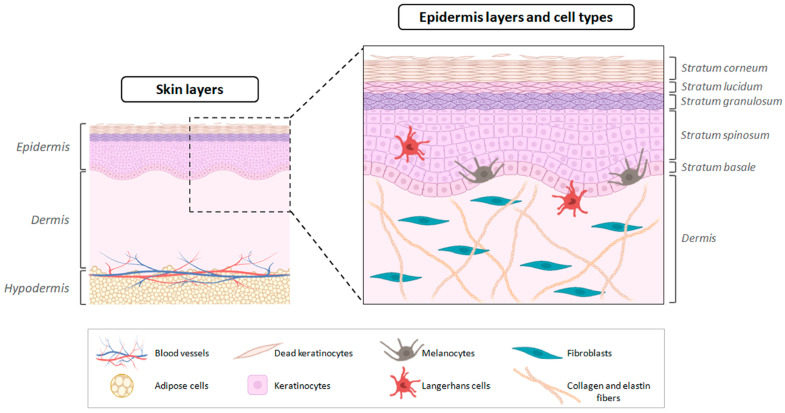
Schematic representation of skin layers (epidermis, dermis and hypodermis) with particular focus on the epidermis: layers (stratum corneum, stratum lucidum, stratum granulosum, stratum spinosum, stratum basale) and most prominent cell types (keratinocytes at different stages, melanocytes, Langerhans cells). Fibroblasts are the primary cell type identified in the dermis.

**Table 1 pharmaceutics-17-01342-t001:** Regulatory acceptance of in vitro skin models and methods in the last three years.

Country/Region	Lead Regulatory Agency(ies)	Validation Agency or Technical Advisor	Accepted Model or Method	Year	Intended Purpose
European Union [40,41,42]	European Medicines Agency (EMA); European Chemicals Agency (ECHA); European Commission (EC)	EURL ECVAM, the European Commission’s center for the validation of alternative methods	IVRT and IVPT in local skin product equivalence dossiers and, update of harmonised lists including biological assessment standards; maintaining alignment with ISO 10993	2024	Supported the demonstration of equivalence of topical formulations using in vitro release and permeation data, integrating barrier and performance assessment, and maintained the acceptance of in vitro skin irritation methods with reconstructed human epidermis via relevant harmonized standards
United Kingdom [45]	Health and Safety Executive (HSE); Medicines and Healthcare products Regulatory Agency (MHRA)	NC3Rs; collaboration with JaCVAM and ICCVAM	OECD TG 497, Defined approaches to skin sensitization	2023	Met skin sensitization requirements, including potency categorization, with integrated batteries that include in vitro testing
United States of America [26,43,44]	Food and Drug Administration (FDA); Environmental Protection Agency (EPA); Consumer Product Safety Commission (CPSC)	NICEATM e ICCVAM	-	2023/ 2025	Significant regulatory advancement for in vitro methods in topicals: The FDA published Draft Guidance for IVPT in ANDAs for topical products in June 2023, which now guides dossiers with in vitro human skin permeation studies. For skin sensitization, the use of defined approaches based on in vitro methods has been consolidated. The EPA updated its strategic vision in 2025, confirming the acceptance of ADs aligned with OECD TG 497, and NICEATM launched new versions of the DASS App in 2024 and 2025 for transparent execution of these ADs.
Canada [48,49]	Health Canada (HC); Environment and Climate Change Canada (ECCC)	SCC and alternative method networks	Inclusion of ISO 10993-23 in the list of recognised standards for medical devices; RHE models	2025	In vitro skin irritation based on reconstructed human epidermis for biological evaluation of medical devices
Brazil [51]	National Health Surveillance Agency (ANVISA)	CONCEA e BRACVAM	-	2023	prohibition of the use of animals when safety and efficacy have already been proven, especially affecting cosmetics.
China [67]	National Medical Products Administration (NMPA)	NIFDC	-	2024	The NIFDC published finalised technical guidelines for cosmetic safety assessment, including a specific guide for integrated skin sensitization testing and assessment strategies using in vitro methods.
India [54]	Central Drugs Standard Control Organization (CDSCO); Bureau of Indian Standards (BIS)	Collaborations with ICCVAM and academic centers	-	-	-
Japan [57]	Pharmaceuticals and Medical Devices Agency (PMDA); Ministry of Health, Labour and Welfare (MHLW)	JaCVAM	Regulatory utility statement of a defined approach to awareness raising	2025	Regulatory use of defined approach combining in vitro methods and other data for skin sensitization
South Korea [55]	Ministry of Food and Drug Safety (MFDS)	KoCVAM	-	-	-
Australia [68]	Therapeutic Goods Administration (TGA); Australian Industrial Chemicals Introduction Scheme (AICIS)	Partnerships with academia and the OECD	Updated categorization guidelines incorporating TG 497	2024	Assessment of skin sensitization following chemical introduction using defined approaches based on in vitro methods

Note: “-“ indicates no changes or acceptance of a new model or method within the past 3 years.

## Data Availability

No new data were created or analyzed in this study. Data sharing is not applicable to this article.

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
