# Peer review of "In Vitro Skin Models as Non-Animal Methods for Dermal Drug Development and Safety Assessment"

_pharmaceutics, 2025, doi:10.3390/pharmaceutics17101342_

Round 1

Reviewer 1 Report

Comments and Suggestions for Authors

The authors have presented a review article focused on non-animal methods used as skin models for safety assessment and dermal drug development of transdermal and topical formulations. The review primarily discusses in vitro skin models, emphasizing their applications in dermatology, cosmetics, pharmaceuticals, and toxicology. Although this work is found to be within the scope of the “Pharmaceutics,” I have the following comments and/or suggestions for improvement:

  1. Please include the main validation and calibration steps involved in developing and assessing skin-on-chip models.
  2. Provide a country-wise overview of these non-animal skin model methods' current regulatory and practical status.
  3. Include a table or summary listing the approved in vitro skin models by different regulatory agencies, along with the year of approval (limited to the last two to three years) and their intended purposes.
  4. The section on pharmaceutical applications would benefit from a more detailed explanation of in vitro-in vivo correction (IVIVC).
  5. The introduction emphasizes the superiority of 3D models, but this is only briefly addressed under the “Advances in Skin Models.”
  6. Consider incorporating an “Expert Opinion” section before the conclusion to provide critical insights or future perspectives.

Author Response

"The authors have presented a review article focused on non-animal methods used as skin models for safety assessment and dermal drug development of transdermal and topical formulations. The review primarily discusses in vitro skin models, emphasizing their applications in dermatology, cosmetics, pharmaceuticals, and toxicology. Although this work is found to be within the scope of the “Pharmaceutics,” I have the following comments and/or suggestions for improvement:"

1. Please include the main validation and calibration steps involved in developing and assessing skin-on-chip models. 

We included three paragraphs in Section 4.1, page 23, describing the main validation and calibration steps for skin-on-chip: verification of the device and perfusion conditions; confirmation of the barrier using complementary metrics (real-time TEER, permeability assays with hydrophilic and lipophilic tracers, histology, and immunostaining of epidermal differentiation); integrity and reproducibility criteria with explicit reporting of electrodes, frequency, temperature, and area; use of reference compounds and formulations for quality control and definition of acceptance ranges; and, when applicable, integration of the data into PBPK models to support clinical predictability.

2. Provide a country-wise overview of these non-animal skin model methods' current regulatory and practical status.

Their suggestions were included in Section 2, page 7, a country-by-country overview of non-animal skin methods and their regulatory and practical status: the EU with widespread access to RHE and a growing focus on IVRT and IVPT; the US with FDA guidelines that consolidate IVPT and barrier readings; the UK with restrictions on animal testing and the adoption of defined approaches; Canada with approval in cosmetics and a shift to reconstructed models; Brazil with regulatory prioritization of alternative methods and recent advances; China with exemptions and technical guidelines for integrated strategies; India and South Korea with restrictions and encouragement of alternatives; and Japan with adoption of OECD guidelines and recognition of defined approaches. We conclude with the summary that, in cosmetics, there is strong convergence toward non-animal methods, while in pharmaceuticals the scenario remains heterogeneous.

3. Include a table or summary listing the approved in vitro skin models by different regulatory agencies, along with the year of approval (limited to the last two to three years) and their intended purposes

We followed the suggestion and included Table 1 in Section 2, page 9. It lists, for the last three years, the in vitro models and methods accepted by different regulatory agencies, indicating the year of acceptance and the intended purpose.

4. The section on pharmaceutical applications would benefit from a more detailedexplanation of in vitro-in vivo correction (IVIVC).

We expanded the pharmaceutical applications section with a straightforward explanation of IVIVC, integrating IVRT and IVPT with sensitive in vivo readings, as well as the use of PBPK models to translate ex vivo measured fluxes and quantities into local and systemic profiles. We also highlighted how choices of skin, diffusion cell, dose, and receptor conditions affect predictive capacity and aligned the discussion with the European regulatory framework for topical product equivalence. We also corrected the typo where it read "IVRT and IVRT," which now reads "IVRT and IVPT."

5. The introduction emphasizes the superiority of 3D models, but this is only briefly addressed under the “Advances in Skin Models.

We appreciate the observation and agree that the argumentation already in the Introduction needs to be reinforced. We have expanded Section 1 (page 4) to concisely explain why 3D models outperform 2D models in preserving the architecture, gradients, and signaling that regulate differentiation, barrier, and inflammatory responses, resulting in more predictable gene expression and pharmacodynamic profiles in humans. We point out that RHE and HSE better reproduce the skin barrier and allow for functional readouts such as TEER and TEWL, in accordance with the RHE method for skin irritation (OECD TG 439). We add that skin-on-a-chip platforms offer controlled perfusion, cocultures, and extended experimental windows, strengthening the in vitro–in vivo correlation. Finally, we situate the topic within the recent regulatory framework, citing the 2024 EMA update and the 2022 FDA position, which consolidate RHE, HSE, and SoC as NAMs and recognize IVRT, IVPT, and barrier integrity metrics such as TEER, TEWL, and ^3H2O.

6. Consider incorporating an “Expert Opinion” section before the conclusion to provide critical insights or future perspectives.

We appreciate the recommendation. In keeping with editorial guidelines and the scope of the manuscript, we have not included the proposed section or additional opinionated content in this version.

Reviewer 2 Report

Comments and Suggestions for Authors

Dear authors

This is an interesting work providing a significant contribution to the field. Some comments on the manuscript

  1. Please use more summarizing tables as the reader can get lost in the information. For example for the cell types used.
  2. The manuscript can significantly benefit from a critical discussion between different available in vitro methods, highlighting validation status, pros and limitations.
  3. The conclusion may include more concrete recommendations for researchers, industry stakeholders, and regulators regarding the adoption and standardization of advanced skin models.
  4. Check the text for abbreviations consistency (eg RhE vs RHE)

Author Response

This is an interesting work providing a significant contribution to the field. Some comments on the manuscript

1. Please use more summarizing tables as the reader can get lost in the information. For example for the cell types used.

Cell types are summarized in Table 2, indicating the model, cell origin, and layers involved. We have also added cross-references in the text to direct the reader to Table 2 when cell types are first mentioned.

2. The manuscript can significantly benefit from a critical discussion between different available in vitro methods, highlighting validation status, pros and limitations.

To address this suggestion, we include in Section 7, page 32, a critical discussion comparing the main in vitro methods, focusing on validation status, strengths, limitations, sources of variability, and context of use. We present a practical framework for model selection based on the study endpoint and regulatory maturity, indicating when RHE, HSE, SoC, IVRT, and IVPT are most appropriate and where evidence gaps persist.

3. The conclusion may include more concrete recommendations for researchers, industry stakeholders, and regulators regarding the adoption and standardization of advanced skin models.

In agreement with the suggestion, we have incorporated in Section 7, page 33, concrete recommendations addressed to researchers, industry, and regulators regarding the adoption and standardization of advanced skin models, including minimum reporting and integrity criteria, best practices for IVPT and barrier metrics, use of reference materials and formulations, harmonization with OECD guidelines, encouragement of interlaboratory studies, and integration of data into PBPK models to support decision-making.

4. Check the text for abbreviations consistency (eg RhE vs RHE)

We reviewed the manuscript to ensure consistency of abbreviations and standardized spelling throughout the text, tables, and captions.

Reviewer 3 Report

Comments and Suggestions for Authors

This review discusses the advancements in in vitro skin models, which are significant, driven by the need for ethical and efficient alternatives to animal testing in the evaluation of cosmetic and pharmaceutical products. It covers the regulations, cells used in reconstructed human skin models, applications, and advancements. While the title mentions “skin-on-chip,” very minimal discussion and articles are covered. Overall, this review highlights the evolution, applications, and future potential of these in vitro models in dermatological research and the cosmetic industry.

In general, the manuscript is well written and documented. Language is easy to understand and cover most important aspects of the topic. List of various skin types in table 1 is very useful.

-There is minimal discussion on skin-on-chip in the review; therefore, I suggest the term can be removed from the title.

However, if you wish to keep the term in the title, then a detailed discussion should be added, discussing various designs on skin-on-chip, how they are used, challenges associated with skin-on-chip models, any throughout assays available with these model, how long it takes to develop these models, etc.

-Table 1: the first column, first row has an issue. Please check and correct. Further, what materials are used for the RHE construction? Any polymer used or specific media used for regeneration can be listed in the table.

Any skin-on-chip covered in table 1? If yes, please specify; if not, please make a separate table for the skin-on-chips models.

-Trans epidermal resistance (TEER) is an important parameter for the RHE skin. In table 1, one column can include the TEER measured/reported in the article. This can be useful information to understand the barrier properties of RHE. Also, please mention  and discuss the TEER standard values for human skin, and RHEs. Likewise, please mention and discuss the other parameters mentioned such as TEWL.

-Discuss the challenges to achieving TEER RHE/RHS model similar to human skin.

-Further, the thickness of RHE and its different layers can be added in Table 1, will be useful information for comparison with human skin thickness.

-It will be useful to discuss 4D skin models in research and commercially available.

For example:

https://www.sciencedirect.com/science/article/pii/S2405886625000028#sec5

https://linkinghub.elsevier.com/retrieve/pii/S1369702117304844

Author Response

This review discusses the advancements in in vitro skin models, which are significant, driven by the need for ethical and efficient alternatives to animal testing in the evaluation of cosmetic and pharmaceutical products. It covers the regulations, cells used in reconstructed human skin models, applications, and advancements. While the title mentions “skin-on-chip,” very minimal discussion and articles are covered. Overall, this review highlights the evolution, applications, and future potential of these in vitro models in dermatological research and the cosmetic industry.

In general, the manuscript is well written and documented. Language is easy to understand and cover most important aspects of the topic. List of various skin types in table 1 is very useful.

1. There is minimal discussion on skin-on-chip in the review; therefore, I suggest the term can be removed from the title.

Thank you for your feedback. We acknowledge that the manuscript addresses skin-on-a-chip only briefly. To align the title with its actual scope—reviewing the main categories of skin models—we decided to remove this term from the title. The new title is: “In Vitro Skin Models as Non-Animal Methods for Dermal Drug Development and Safety Assessment.

2. Table 1: the first column, first row has an issue. Please check and correct. Further, what materials are used for the RHE construction? Any polymer used or specific media used for regeneration can be listed in the table.

We corrected the issue in the first cell of Table 2. Regarding the materials used in RHE construction, there is significant heterogeneity between protocols regarding medium and supplementation, coating matrix, cell sources, and polymers. Many articles do not report these items completely, making a comparable and up-to-date list impossible in this review. To avoid omissions or inconsistencies, we kept Table 2 focused on models, cell types, functional aspects, and regulatory acceptance. We propose addressing materials and formulations in a future work dedicated to this topic.

3. Any skin-on-chip covered in table 1? If yes, please specify; if not, please make a separate table for the skin-on-chips models.

To address the suggestion, we chose to include skin-on-a-chip entries directly in Table 2, specifying the platforms and their purposes, to keep the regulatory landscape integrated and consistent with the scope of the review, rather than creating a dedicated SoC table. We felt that a separate table would shift the focus of the article. Additionally, we incorporated one of the articles indicated by the reviewer in the last row of Table 2, as requested.

4. Trans epidermal resistance (TEER) is an important parameter for the RHE skin. In table 1, one column can include the TEER measured/reported in the article. This can be useful information to understand the barrier properties of RHE. Also, please mention  and discuss the TEER standard values for human skin, and RHEs. Likewise, please mention and discuss the other parameters mentioned such as TEWL.

To address this suggestion, instead of including RHE thickness in Table 2, we added four paragraphs in Section 4.1, pages 22–24, clarifying why these values ​​are not uniformly comparable: TEER and thickness vary with temperature, geometry and effective area, electrode placement and frequency, and structural differences between RHE/RHS and ex vivo skin that affect conduction and response to perturbations. We also contextualize that, in toxicological and cosmetic practice, regulatory acceptance of RHE methods is anchored in performance standards and primary functional readouts, such as MTT and ET50 with appropriate controls, complemented by immunostains, so TEER remains a useful but method- and context-dependent metric. We retained Table 2's models, cell types, and function, while the discussion of thickness and integrity is detailed in the text.

5. Discuss the challenges to achieving TEER RHE/RHS model similar to human skin.

We appreciate the suggestion. The challenges of achieving TEER values ​​in RHE and RHS comparable to human skin were discussed in Section 4.1, page 24.

6. Further, the thickness of RHE and its different layers can be added in Table 1, will be useful information for comparison with human skin thickness.

We recognize the value of comparing RHE thickness to that of human skin, but the available data are reported heterogeneously across platforms and batches. Therefore, rather than including potentially inconsistent values ​​in the former Table 1 (now Table 2), we added a paragraph in Section 4, page 21, clarifying this variability and summarizing how commercial RHEs often emphasize histological similarity without standardizing thicknesses, how full-thickness models describe layers with scale bars not always converted to mean micrometers, and how in skin-on-a-chip and 3D bioprinting, thickness depends on experimental parameters.

7. It will be useful to discuss 4D skin models in research and commercially available.

We agree with the importance of this suggestion and, therefore, included in Section 6, page 31, a discussion of 4D skin models, covering the concept, applications in wound healing, infection control, and microfluidic-integrated dynamic barriers, as well as the main responsive materials. We note that commercial availability is still incipient compared to RHE and full-thickness models, with challenges of standardization, durability, long-term biocompatibility, and regulatory validation. The suggested references were included in the review.
